# Shifts in myeloarchitecture characterise adolescent development of cortical gradients

Casey Paquola[1†*], Richard AI Bethlehem[2,3†*], Jakob Seidlitz[2,4], Konrad Wagstyl[2], Rafael Romero-Garcia[2], Kirstie J Whitaker[2,5], Reinder Vos de Wael[1], Guy B Williams[6], NSPN Consortium, Petra E Vértes[2,5], Daniel S Margulies[7], Boris Bernhardt[1†*], Edward T Bullmore[2,6†*]

[1]Multimodal Imaging and Connectome Analysis Lab, McConnell Brain Imaging Centre, Montreal Neurological Institute and Hospital, McGill University, Montreal, Canada; [2]Department of Psychiatry, University of Cambridge, Cambridge, United Kingdom; [3]Autism Research Centre, Department of Psychiatry, University of Cambridge, Cambridge, United Kingdom; [4]Developmental Neurogenomics Unit, National Institute of Mental Health, Bethesda, United States; [5]The Alan Turing Institute, London, United Kingdom; [6]Department of Clinical Neurosciences, Wolfson Brain Imaging Centre, University of Cambridge, Cambridge, United Kingdom; [7]Frontlab, Institut du Cerveau et de la Moelle épinière, UPMC UMRS 1127, Inserm U 1127, CNRS UMR 7225, Paris, France

*For correspondence:
casey.paquola@gmail.com (CP);
rb643@medschl.cam.ac.uk (RAIB);
boris.bernhardt@mcgill.ca (BB);
etb23@cam.ac.uk (ETB)

†These authors contributed equally to this work

Competing interests: The authors declare that no competing interests exist.

**Abstract** We studied an accelerated longitudinal cohort of adolescents and young adults (n = 234, two time points) to investigate dynamic reconfigurations in myeloarchitecture. Intracortical profiles were generated using magnetization transfer (MT) data, a myelin-sensitive magnetic resonance imaging contrast. Mixed-effect models of depth specific intracortical profiles demonstrated two separate processes i) overall increases in MT, and ii) flattening of the MT profile related to enhanced signal in mid-to-deeper layers, especially in heteromodal and unimodal association cortices. This development was independent of morphological changes. Enhanced MT in mid-to-deeper layers was found to spatially co-localise specifically with gene expression markers of oligodendrocytes. Interregional covariance analysis revealed that these intracortical changes contributed to a gradual differentiation of higher-order from lower-order systems. Depth-dependent trajectories of intracortical myeloarchitectural development contribute to the maturation of structural hierarchies in the human neocortex, providing a model for adolescent development that bridges microstructural and macroscopic scales of brain organisation.
DOI: https://doi.org/10.7554/eLife.50482.001

## Introduction

Adolescence is a crucial phase in biological and psychosocial maturation and involves large-scale reconfigurations of brain anatomy (*Paus et al., 2008*). Previous histopathological and neuroimaging studies have shown marked age-related changes in brain structure during this sensitive period (*Giedd et al., 2015*; *Raznahan et al., 2011*; *Sowell et al., 2004*). This is particularly evident in magnetic resonance imaging (MRI) assessments of the macro-structural morphology of cortical regions, which have revealed regionally-variable dynamics of cortical thinning in adolescence (*Giedd et al., 2015*; *Raznahan et al., 2011*; *Sowell et al., 2004*). While these findings confirm the existence of strong biological forces shaping adolescent brain anatomy, morphometric analyses typically only

quantify shape changes of the inner and outer cortical boundaries. In turn, these analyses may not be specific for microstructural changes occurring within the cortical mantle, which ultimately play key roles in adolescent development of cortical connectivity and function (*Huntenburg et al., 2017*). Several recent neuroimaging studies assessed intracortical microstructure in adolescence. One promising imaging technique is magnetisation transfer (MT), an MRI acquisition sequence that is sensitive to how water molecules interact with macromolecules in the brain, notably myelin (*Schmierer et al., 2007*). Although techniques such as MT cannot be equated with cortical myelin content per se (*Serres et al., 2009a*; *Serres et al., 2009b*), the MT parameter is dominated by myelin-related molecules making this technique a viable in vivo proxy for the contrast seen histologically in myelin basic protein (*Koenig, 1991*; *Odrobina et al., 2005*; *Whitaker et al., 2016*). A post mortem study in patients with multiple sclerosis has also shown that MT measures scale with demyelination and remyelination, suggesting dependence of this measure on myelin content (*Schmierer et al., 2007*). While correlated with alternative myelin-sensitive imaging, MT is arguably the strongest in vivo marker of myelin, based on its spatial correspondence with myelin basic protein (*Whitaker et al., 2016*). While recent studies confirmed associations between adolescence and changes in myelin-markers (*Carey et al., 2018*; *Grydeland et al., 2019*; *Kwon et al., 2018*; *Whitaker et al., 2016*), findings have been mainly cross-sectional, precluding inferences on intra-individual trajectories. Furthermore, studies have generally focused on specific depths or intracortical averages, ignoring depth-dependent dynamics and thus not addressing potential systematic shifts in cortical myeloarchitecture and lamination in adolescence.

Quantitative profiling of intracortical properties across cortical depths, and specifically parameterization using central moments (*Papoulis and Unnikrishna Pillai, 2002*), has been proposed to characterise cytoarchitecture in seminal histological work (*Schleicher et al., 1999*) and capture inter-individual variation (*Amunts et al., 1999*). In essence, studying the mean (first moment) of MT profiles perpendicular to the cortical mantle allows inferences on overall myelin content while higher order moments can address depth-dependent changes (*Figure 1*). Analysis of skewness (third moment) can contrast relative properties of deep and superficial cortical layers, and depth is a critical dimension of laminar differentiation that relates to architectural complexity (*Zilles et al., 2002*) and cortical hierarchy (*Mesulam, 1998*). Applied to adolescence, such an analysis offers a non-invasive window into cortical architecture, which may recapitulate and expand classical histological findings showing overall increases in cortical myelin in adolescence (*Kaes, 1907*) as well as transcriptional analyses suggesting that laminar signatures reflect cortical development (*Miller et al., 2014*). In addition to studying regional variations in cortical architecture, depth-dependent profiling theoretically lends a framework to tap into the large-scale coordination of different brain areas. One such approach, known as microstructure profile covariance (MPC), quantifies inter-regional similarity in microstructure at a single subject-level (*Paquola et al., 2019*). Previous research has demonstrated the utility of MPC for understanding large-scale patterns of cortical architecture, specifically illustrating a sensory-fugal gradient of microstructural differentiation in both post mortem and in vivo datasets (*Paquola et al., 2019*). This axis describes gradual transitions from primary sensory and motor regions with high laminar differentiation, through association cortex toward paralimbic areas with increasingly dysgranular appearance (*Mesulam, 1998*). Given prior evidence that microstructural similarity predicts cortico-cortical connectivity (*Barbas and Rempel-Clower, 1997*), tracking age-related changes in MPC provides an unprecedented way to probe coordinated maturation of microstructural networks during the critical adolescent period, moving towards a network perspective of structural brain development (*Alexander-Bloch et al., 2013*; *Váša et al., 2018*).

The present study examined adolescent maturation of cortical architecture and microstructure using an accelerated longitudinal NeuroScience in Psychiatry Network dataset (*Kiddle et al., 2018*; 234 participants scanned twice). We translated quantitative profile analysis via statistical moment parameterisation, previously proposed for histological data, to surface-based MT imaging data and cross-referenced findings against established atlases of cytoarchitectural complexity and laminar differentiation (*Von Economo and Koskinas, 2008*). We tracked longitudinal changes in MT profile moments using linear mixed effect models, which leverage subject- and group-level effects to estimate microstructural changes across the entire age range (*Carey et al., 2018*). Based on histological work (*Palomero-Gallagher and Zilles, 2019*; *Zilles et al., 2002*), we hypothesized that specifically the first (mean) and third (skewness) central moments of the MT profiles would capture different biological mechanisms and exhibit divergent maturational trajectories. The mean was expected to

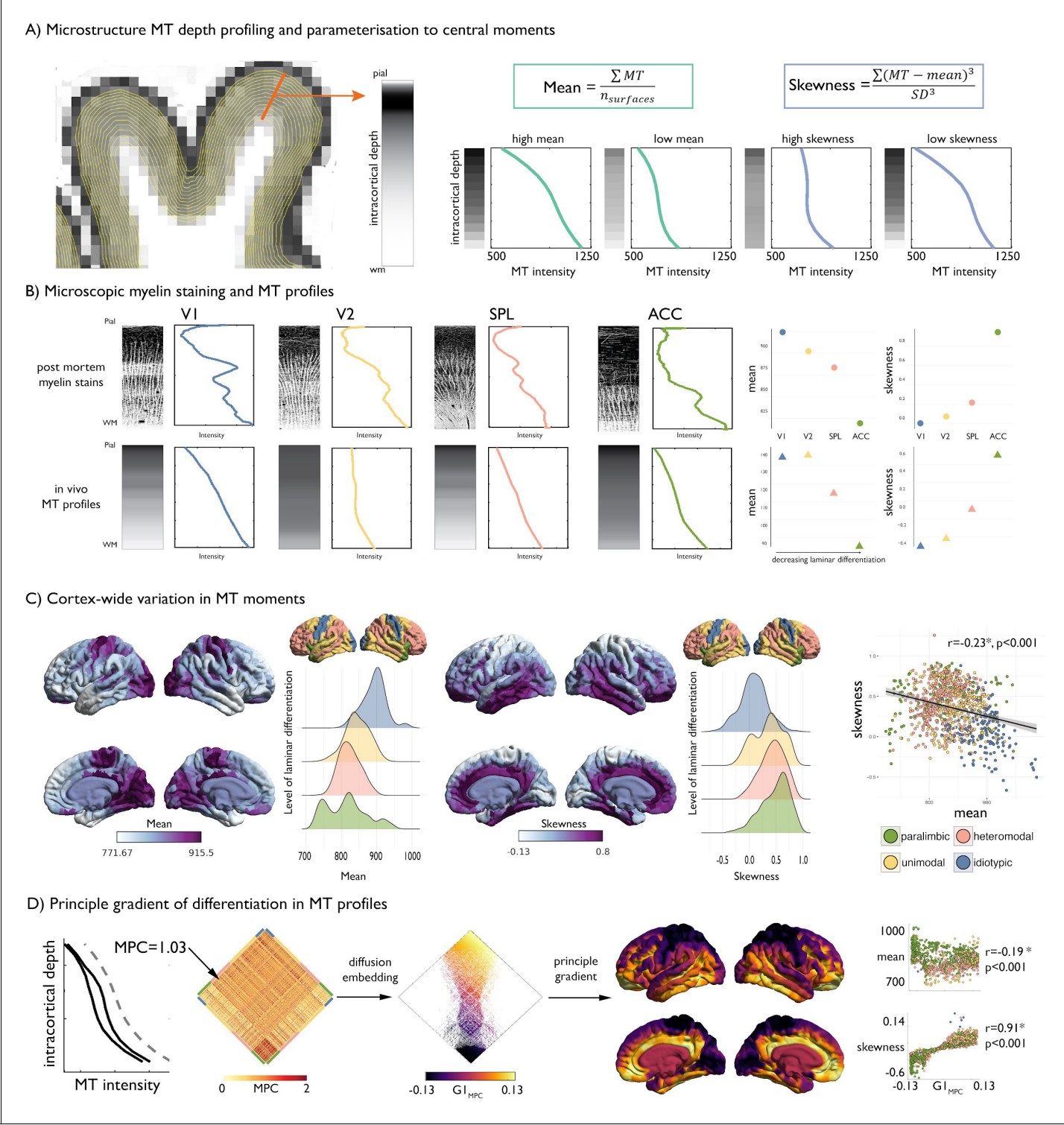

**Figure 1.** Intracortical MT depth profiling. (A) *Left.* Equivolumetric surfaces overlaid on an MT image of a single subject, also showing an example vertex along which MT intensity is sampled (example MT profile in grey, with lighter tones represent higher MT intensity). *Right.* Schema of first and third moments. (B) *Left.* Examples of microscopic myelin-stained sections (inverted image shows myelin in lighter tones) (***Vogt and Vogt, 1919***; ***Vogt, 1911***), with corresponding profiles and group-average MT profile from the same regions extracted in vivo. V1 = primary visual cortex. V2 = secondary visual area. SPL = superior parietal lobule. ACC = anterior cingulate cortex. *Right.* Dot-plots showing the moments for each exemplar profile. (C) Baseline group-average of first and third moment plotted on the cortical surface, and within levels of laminar differentiation (***Mesulam, 2002***; ***Paquola et al., 2019***). The scatterplot shows the spatial overlap between first and third moment, coloured by level of laminar

*Figure 1 continued on next page*

*Figure 1 continued*

differentiation. Findings on the second and fourth moment are shown in *Figure 1—figure supplement 1*, and with an alternative cytoarchitectural taxonomy in *Figure 1—figure supplement 2*. (D) Microstructure profile covariance (MPC) was estimated between each pair of nodes based on the partial correlation of two nodes' MT profiles (*black*), controlling for the cortex-wide mean intensity profile (*grey dashed*). Baseline MPC matrices were averaged across the group, and diffusion map embedding was employed to order regions according to the principal gradient of microstructural differentiation (G1$_{MPC}$). Scatterplots show the relationship between node-wise loadings onto the principle gradient with MT moments.

DOI: https://doi.org/10.7554/eLife.50482.002

The following figure supplements are available for figure 1:

**Figure supplement 1.** Baseline properties of SD and kurtosis.

DOI: https://doi.org/10.7554/eLife.50482.003

**Figure supplement 2.** Replication with different classification of cytoarchitectural complexity (*Von Economo and Koskinas, 2008*).

DOI: https://doi.org/10.7554/eLife.50482.004

largely encompass similar changes to those captured by previous in vivo imaging studies on overall MT changes (*Carey et al., 2018*; *Grydeland et al., 2019*; *Kwon et al., 2018*; *Whitaker et al., 2016*) corresponding to a shift in the overall cortical myelin content. Conversely, we anticipated skewness would capture shifts of MT intensity in the depth dimension of the intensity profile. Intracortical MT profile analysis was complemented by a series of cortical thickness and transcriptomic overlap analyses to assess morphological correlates and molecular underpinnings. In addition to studying regional MT profiles, we leveraged the MPC framework to tap into cortex-wide coordination of adolescent MT profile changes between different brain areas and consolidate our findings at a system level.

## Results

### Characterization of intracortical MT profiles

We studied the NSPN dataset, an accelerated longitudinal cohort that aggregates multimodal imaging data from 234 adolescents and young adults aged 14–27 (for details on cohort selection, processing, and quality control, see Materials and methods). Briefly, we generated cortical surface models based on T1-weighted MRI and co-registered the corresponding MT volumes to these surfaces. We systematically generated equivolumetric intracortical surfaces and sampled MT intensities along matched vertices perpendicular to the cortical mantle to build intracortical MT profiles (*Figure 1A*). Our vertex-wise technique leverages equivolumetric transformations that critically adjust surface placement in accordance to the folded cortical sheet (*Wagstyl et al., 2018*), thereby better coinciding with the position of the putative cortical laminae than equidistant or Laplace-field guided techniques (*Waehnert et al., 2014*). Our intracortical approach thus offered better precision and biological validity over conventional voxel-based morphometry techniques that may be agnostic to cortical topology and intracortical architecture, and that may amplify mixing of different tissue types (*Ziegler et al., 2019*). MT profiles at each vertex were parcellated in 1012 approximately equal sized nodes and parameterized via central statistical moments (*Zilles et al., 2002*). We focused on the first moment (mean across all cortical depths) and third moment (skewness across cortical depths), which are readily interpretable in terms of, respectively, mean myelin content and the ratio of myelination in deeper compared to more superficial layers; and have been studied in prior histological work (*Zilles et al., 2002*).

### Baseline MT profiles, moments and covariance

At baseline (*i.e.*, using the average across timepoint 1), the cortex-wide average MT profile shows a non-linear increase in intensity from the superficial layers, adjacent to the pial boundary, towards the deeper layers approaching the white matter boundary, consistent with prior literature (*Whitaker et al., 2016*; *Figure 1A*). We first assessed the correspondence of intracortical MT profiles with myeloarchitecture by comparing MT profiles with histological myelin stains from four regions of interest, representing the four levels of laminar differentiation (*Mesulam, 2002*; *Palomero-Gallagher and Zilles, 2019*; *Vogt and Vogt, 1919*; *Vogt, 1911*; *Figure 1B*, see Materials and methods for details on profile quantification). Although post mortem myelin stained sections and in vivo MT profiles differ in terms of resolution and specificity to myelin, we observed

similar variations in mean and skewness of profiles across levels of laminar differentiation (*Figure 1B*), supporting the extension of profile analysis from histology to in vivo MT imaging. At a whole cortex-level, mean MT was highest in idiotypic cortex and decreased with less laminar differentiation, while skewness exhibited an opposite pattern (spatial correlation = −0.23, $p_{spin}$ <0.001; *Figure 1C*, *Figure 1—figure supplements 1–2*, *Appendix 1—figure 1*). Negative or near-zero skewness was observed in idiotypic and unimodal areas, relating to more even distribution of MT across cortical depths, whereas more positive skewness was observed in heteromodal and paralimbic areas, related to higher MT intensities in infragranular compared to supragranular layers (*Zilles et al., 2002*; *Figure 1C*).

We explored the topology of intracortical MT patterns using microstructure profile covariance (MPC). By correlating depth-wise MT profiles, the MPC procedure estimates inter-regional microstructural similarity and has previously been validated using a combination of post mortem and in vivo data (*Paquola et al., 2019*; *Figure 1D*). Diffusion map embedding, a nonlinear dimensionality reduction technique, was employed to resolve the principal axis of microstructural differentiation. The relative positioning of nodes in this embedding space informs on similarity of their covariance patterns. In line with previous work (*Paquola et al., 2019*), the first principal gradient within the baseline cohort was anchored on one end by idiotypic sensory regions and by paralimbic regions on the other end. This sensory-fugal gradient reflects systematic variations in the MT profiles; it was strongly correlated with MT profile skewness (r = 0.91, $p_{spin}$ <0.001; *Figure 1D*) and weakly with mean MT (r = 0.19, $p_{spin}$ <0.001).

## Age-related changes in cortical depth MT profiles

Studying age-related changes in MT moments, we observed that adolescence led to a significant increase in the mean ($q_{FDR}$ <0.00625, % rate of change: [4.8 14.7] 95% CIs; *Figure 2*, *Figure 2—figure supplement 1*), suggesting overall increases in intracortical myelin content. Conversely, we observed a unique spatial pattern of decreases in skewness ($q_{FDR}$ <0.00625, % rate of change: [−179.2–34.6] 95% CIs; *Figure 2*). Depth-wise changes in MT intensity suggest that these decreases in skewness reflected a preferential increases in MT in mid to deeper layers, suggesting accumulation in myelin. Controlling for overall mean MT trajectories resulted in virtually identical patterns of skewness change, suggesting relative independence between skewness and mean MT trajectories in adolescence (*Appendix 1—figures 2–4*). The differential impact of age across levels of laminar differentiation was subsequently assessed by spin permutations (*Alexander-Bloch et al., 2018*). Across all moments, age-related changes in skewness were preferentially located in heteromodal (z > 3.06, p<0.043) and unimodal cortex (z > 3.24, p<0.024), while idiotypic nodes were less prominently represented than expected by chance (*Figure 2—figure supplement 1* and *Supplementary file 1*).

## Independence of age-related intracortical MT profiles changes from cortical morphology

To examine the specificity of MT profile trajectories to intracortical variations, we assessed robustness of findings against changes in cortical morphology and boundary definition (*Bernhardt et al., 2018*; *Whitaker et al., 2016*). To this end, we computed residual MT moments by controlling for thickness or interface blurring at each vertex within node-wise linear models. The spatial distribution of MT profile mean and skewness were virtually unchanged when data were additionally controlled for morphological and intensity confounds (r > 0.99; *Appendix 1—figures 2–4*). Age-related changes in MT moments were also virtually identical when using MT profile data corrected for cortical thickness and interface blurring (r > 0.98, *Appendix 1—figures 2–4*), supporting that MT profile trajectories were primarily driven by intracortical factors. Furthermore, spatial correlations suggested that MT moments were relatively independent of regional variations in laminar thickness derived from a post mortem volumetric reconstruction of a Merker-stained human brain (*Wagstyl et al., 2018*) (mean: −0.06 < r < 0.05. skewness: −0.17 < r < 0.02; *Appendix 1—figure 5*).

## Age-related change in MT profiles co-localised with expression of oligodendroglial genes

To explore molecular substrates of our imaging results, we referenced our findings against post mortem gene expression maps provided by the Allen Institute for Brain Sciences using Neurovault API

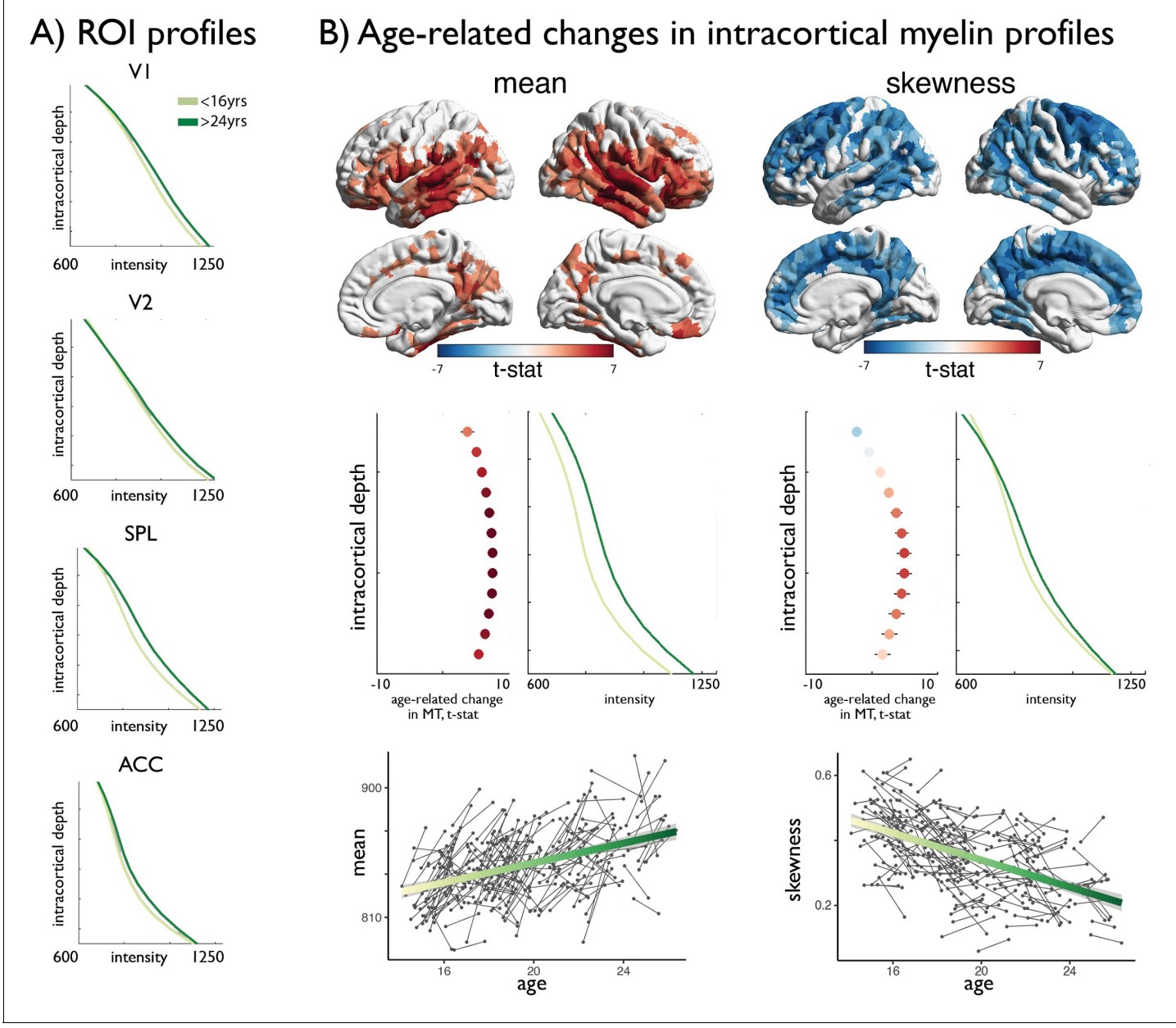

**Figure 2.** Age-related changes in MT moments. (A) Shifts in MT profiles and moments from lowest to highest age strata shown for exemplar regions of laminar differentiation (*Figure 1*). (B) *Upper.* Age-related changes in MT moments (qFDR <0.00625). *Middle.* t-statistic (mean ± SD) of age-related changes in MT intensity at each intracortical surface across significant regions. Mean increases were balanced across surfaces, whereas decreases in skewness were driven by preferential intensity increases at mid-to-deeper surfaces. *Lower.* Individual changes across significant regions, with regression lines depicting age-related changes across the investigated range.

DOI: https://doi.org/10.7554/eLife.50482.005

The following figure supplement is available for figure 2:

**Figure supplement 1.** Age-related changes in SD and kurtosis.

DOI: https://doi.org/10.7554/eLife.50482.006

gene decoding (*Gorgolewski et al., 2015*; *Gorgolewski et al., 2014*; *Hawrylycz et al., 2012*). We identified genes whose expression pattern spatially resembled the maps of the age-related change from our in vivo MT analysis (*Figure 2B*; *Hawrylycz et al., 2012*). Including only those genes that passed multiple comparisons corrections ($p_{FDR}$ <0.05), we conducted enrichment analysis of

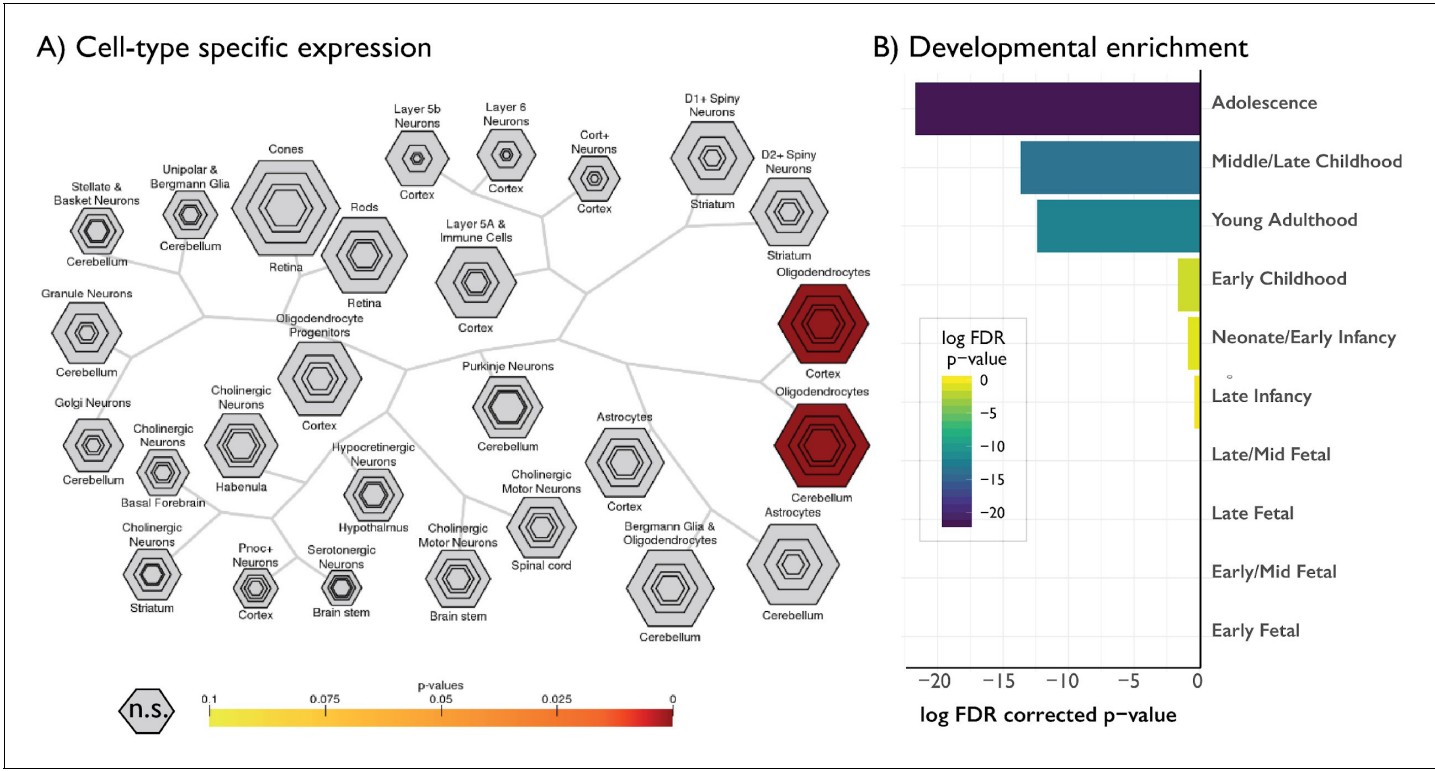

**Figure 3.** Gene decoding of age-related changes in skewness against the Allen Institute for Brain Sciences gene expression atlas. Only genes negatively relating to age-related changes in skewness are shown as only these genes survived FDR < 0.05. No other moments had any significantly associated genes. Cell-type-specific expression, showing selective enrichment for oligodendrocytes, reinforcing the link between profile skewness and myelination. Hexagon rings denote significance at different pSI thresholds (from p<0.05 in the outer ring to 0.00001 in the centre). (B) Developmental cortical enrichment, showing enrichment specifically in adolescence and young adulthood.

DOI: https://doi.org/10.7554/eLife.50482.007

The following figure supplement is available for figure 3:

**Figure supplement 1.** Gene ontology analyses from baseline profiles of the four moments of the intensity distribution.

DOI: https://doi.org/10.7554/eLife.50482.008

standard mammalian phenotypes (*Smith and Eppig, 2012*), cell-specific expression analysis and developmental expression analysis across developmental time windows (*Dougherty et al., 2010*).

The pattern of age-related change in skewness showed a significant and specific transcriptomic signature (for unthresholded lists, see *Data Availability*). Cell specific expression analysis (*Dougherty et al., 2010*) suggested enrichment exclusively with oligodendrocytes (p<0.001 [at a specificity index threshold of 0.0001]; *Figure 3A*), confirming the association between myelination and decreased MT skewness. Developmental expression analysis (*Dougherty et al., 2010*) showed selective enrichment for adolescence and young adulthood (*Figure 3B*). Both analyses thus confirmed a spatial overlap of our findings from NSPN with genes associated with myelin processes, that are also enriched during adolescence. It should be noted however that individual gene expression assays only provide a snapshot of inherently dynamic processes (*Arnatkevic̆iūtė et al., 2019*). We detected enrichment of adolescent-linked genes based on cross-sectional developmental gene expression data. Specifically, genes showing significant spatial overlap in adult post-mortem brains from the AIBS were identical to those showing significant expression differences during adolescence in the BrainSpan dataset. While there was evidence for an adolescent developmental signal results are nevertheless indirect, also given that glial and oligodendrocyte associated genes may undergo expression changes during other periods of the lifespan (*Soreq et al., 2017*). Furthermore, we performed gene ontology analysis using a modified Fishers exact test, which captures the deviation from the expected gene rank (*Kuleshov et al., 2016*), on genes associated with decreasing skewness. This indicated a negative association with demyelination (Z = −2.16, p=0.001) and axon

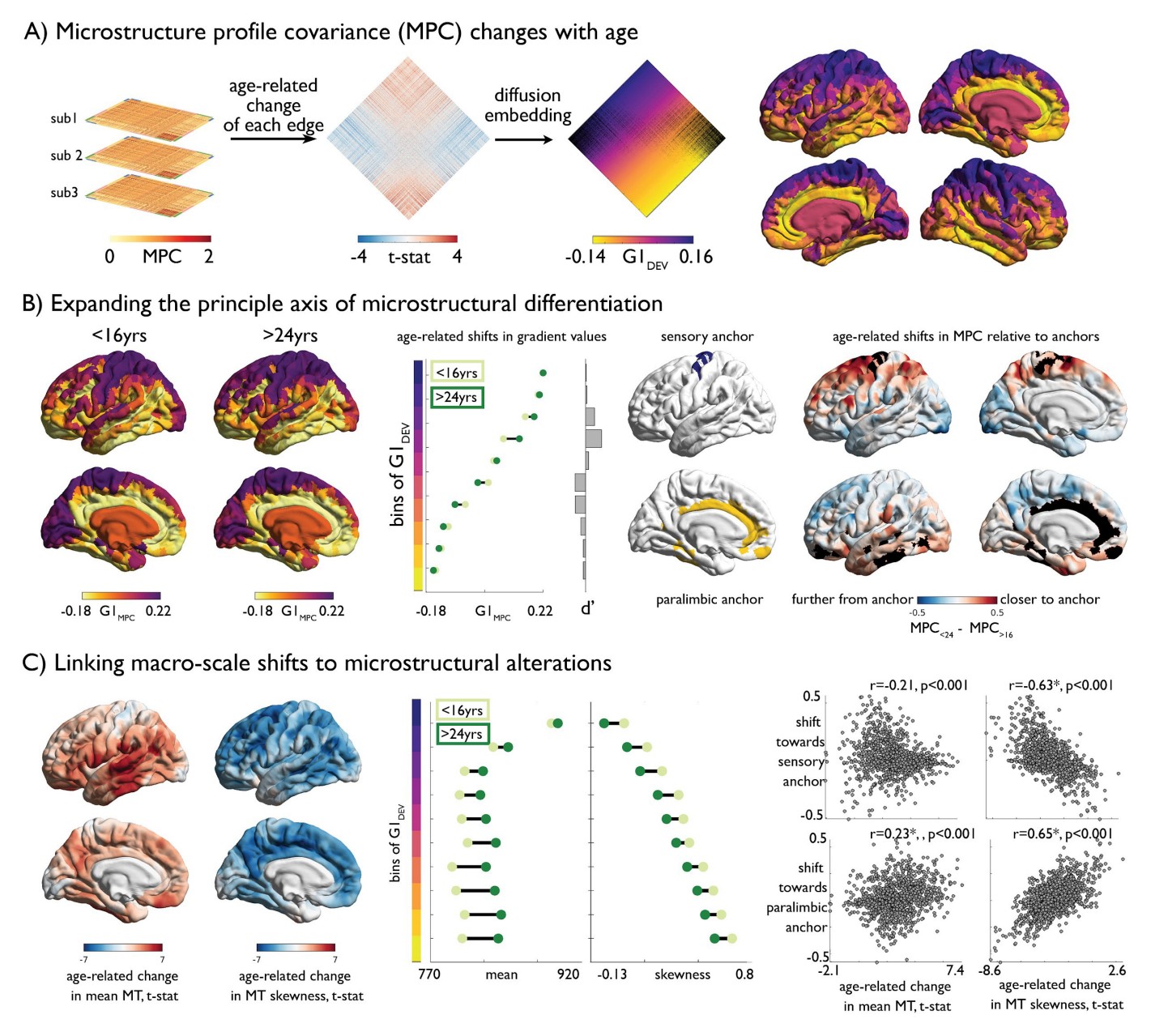

**Figure 4.** Age-related changes in microstructure profile covariance (MPC). (A) Subject-specific MPC matrices (*stacked*) were used in mixed effect models to calculate age-related changes in microstructural similarity between each node pair, generating a t-statistic matrix (*middle*). Diffusion map embedding ordered this matrix along the principal axis (G1_DEV) (*right*). Rows of the matrix were coloured according to G1_DEV. Surface projection of G1_DEV illustrates a transition from primary sensory (*purple*) through association (*pink*) to limbic cortices (*yellow*). (B) *Left.* Principal axis of microstructural differentiation within extreme age strata (G1_MPC), show a sensory-fugal axis similar to the baseline and developmental analysis. *Middle.* Shifts in average G1_MPC within ten discrete bins of G1_DEV, and corresponding Cohen's d effect size. Central regions of G1_DEV, aligning with association cortices, expand from the centre of G1_MPC, towards either sensory or paralimbic anchors. *Right.* Age-related shifts towards anchors visualised via age-strata difference in MPC. The average MPC to each anchor was calculated within each extreme age-strata then subtracted. Prefrontal and parietal areas increased in microstructural similarity with the sensory anchor (*top*), whereas temporal regions became more coupled with the paralimbic anchor. (C) *Left.* Age-related change in MT moments (unthresholded maps from *Figure 2B*). *Middle.* Age-related shifts in mean and skewness within each bin of the gradient. *Right.* Correlation betweeBn age-related change in MT moments (*Figure 2B*) and shifts to anchors (B *right*).
DOI: https://doi.org/10.7554/eLife.50482.009

The following figure supplements are available for figure 4:

**Figure supplement 1.** Deconstructing age-related changes in MPC.

*Figure 4 continued on next page*

*Figure 4 continued*

DOI: https://doi.org/10.7554/eLife.50482.010

**Figure supplement 2.** Increasing bimodality of the principle gradient of microstructural differentiation.

DOI: https://doi.org/10.7554/eLife.50482.011

**Figure supplement 3.** Developmental gradients derived from moment difference.

DOI: https://doi.org/10.7554/eLife.50482.012

degeneration (Z = −2.34, p=0.011), reinforcing associations to myelin. In sum, genes negatively associated with age-related skewness reduction were less likely to be enriched for demyelination and were exclusively associated with oligodendrocytes. Given that the present data are based on healthy cortex, those same areas are not high in expression of 'demyelinating' genes, but instead show a negative signal (*e.g.*, negative z-scores). This is expected since those areas of high expressing myelin genes are most vulnerable (*Seidlitz et al., 2019*). Thus, MT skewness changes during adolescent development appear to be strongly reflective of changes in myelination. None of the other moments provided a list of genes passing the threshold set by the false discovery rate procedure. However, exploratory rank-based correlations on the weights of each gene indicated a small correlation between mean and skewness related genes (rho = 0.14, p<0.001), indicating mean and skewness share a modest transcriptomic signature. Finally, it should be noted that the same analysis on baseline data only (average across a single scan per subject) from this cohort did not identify a whole genome transcriptional profile that overlapped with any of the baseline MT moments (*Figure 3—figure supplement 1*), reinforcing the added value of developmental data (e.g. measuring an age-related change) in detecting human neurodevelopmental processes by myelin-sensitive MRI.

## Adolescent reconfiguration of microstructure profile covariance networks

Adopting the MPC framework, we subsequently assessed inter-regional coordination of myeloarchitectural development. MPC networks were constructed for each participant, and we modelled age-related changes in microstructural similarity between all node pairs (*Figure 4A*). At the node-to-node edge-level, adolescent development was primarily related to increases in MPC (1564 increases vs 534 decreases at $q_{FDR}$ <0.025; *Figure 4—figure supplement 1*). To express the spatial pattern of myeloarchitectural development in a lower dimensional, and more readily interpretable space, we implemented diffusion map embedding (*Margulies et al., 2016*; *Figure 4A*). Nodes closer in this embedding space have more similar patterns of age-related change in MPC, whereas distant nodes undergo dissimilar development. The first principal component ($G1_{DEV}$) illustrated a sensory-fugal gradient, explaining 33% variance (*Figure 4A*). On one end of the gradient, idiotypic sensory and motor areas became increasingly coupled in adolescence, and more segregated from the opposite anchor constituted mainly of paralimbic nodes. The concordance of the developmental gradient ($G1_{DEV}$) with the baseline MPC gradient (r = 0.89, p<0.001, *Figure 1D*) suggested that the axis of microstructural differentiation expands during adolescence. To synoptically visualise these dimensional changes, we generated, aligned, and contrasted cross-sectional MPC gradients ($G1_{MPC}$) within the youngest (<16 years) and oldest (>24 years) age strata. Similar results were observed using alternative age windows. Heteromodal and unimodal cortex, occupying central regions of the gradient, were drawn outwards towards one of the gradient anchors in older individuals (*Figure 4B*), forming a more bimodal distribution (*Figure 4—figure supplement 2*). Specifically, prefrontal and medial parietal areas were increasingly coupled with sensory areas, and temporal regions increased in similarity to paralimbic regions (*Figure 4B*). By relating the age-related shifts in MPC to anchors with age-related changes in MT moments, we found that greater decreases in MT skewness were related to sensory anchor coupling (r = −0.63, $p_{spin}$ <0.001; *Figure 4C*; *Figure 4—figure supplement 3*). Conversely, restricted age-related skewness changes were associated with paralimbic anchor coupling (r = 0.65, $p_{spin}$ <0.001). Together, these findings demonstrate how depth-dependent myelination during adolescence reshapes myeloarchitecture and underpins macro-scale reorganisation of cortical networks.

## Discussion

Our longitudinal analyses revealed marked alterations in intracortical myeloarchitecture during adolescence, which was accompanied by large-scale reorganisation of microstructural gradients. Changes to intracortical myelin can be generally characterised by two developmental processes, one involving overall increases in mean MT and one involving preferential MT increases in mid-to-deeper cortical layers. Both processes particularly affected unimodal and heteromodal association cortices that is areas between low-level sensory and motor areas on the one end of the cortical hierarchy and high-level transmodal and limbic regions on the other end. A series of complementary explorations, based on alternative neuroimaging features known to undergo large-scale changes in adolescence (*Giedd et al., 2015*; *Raznahan et al., 2011*; *Sowell et al., 2004*), demonstrated the independence of these intracortical microstructural changes from age-related effects on overall cortical morphology and tissue contrast. In addition, cross-referencing our findings against post mortem gene expression maps showed that patterns of age-related change overlapped with genes associated with oligodendrocyte processes, supporting the specificity of MT changes to myelin. Leveraging microstructure profile covariance analysis, we explored coordination of intracortical microstructure characteristics across distributed areas (*Paquola et al., 2019*). This showed that local processes were paralleled by a cortex-wide differentiation along the sensory-fugal axis. This axis ultimately underpins the segregation of lower- from higher-order components of the putative cortical hierarchy.

Inspired by classical histological studies operating on 2D sections (*Amunts et al., 1999*; *Schleicher et al., 1999*; *Zilles et al., 2002*), our work formulated a surface-based approach to quantify myeloarchitecture in vivo via systematic profiling of intracortical MT intensities in the direction of cortical columns and its parameterization via the profiles' central moments. The skewness of the MT profile, reflecting nonlinear increases in myelin towards deeper layers (*Dinse et al., 2015*; *Sprooten et al., 2019*; *Waehnert et al., 2014*), tilts from infragranular-dominant in paralimbic regions towards a more even distribution across cortical depths in sensory regions. This graded shift parallels changes in the laminar origin of projections, where paralimbic regions engaged in feedback processing project from infragranular layers and sensory regions providing feedforward information project predominantly from supragranular layers (*Felleman and Van Essen, 1991*). Meanwhile, mean myelin increased from polar towards sensory regions, recapitulating prior post mortem evidence (*Vogt and Vogt, 1919*; *Vogt, 1911*), established atlases of cytoarchitecture (*Von Economo and Koskinas, 2008*), and resting-state functional connectivity gradients (*Hong et al., 2017*; *Margulies et al., 2016*). These baseline analyses demonstrate the power of this framework to profile human myeloarchitecture in vivo.

Tracking longitudinal change in intracortical MT profiles, we found concomitant but independent developmental patterns in the trajectory of mean and skewness. Both sets of age-related changes were strongly and preferentially located in heteromodal and unimodal association cortex; yet, they only partially overlapped, a finding also supported by gene expression analysis. Only skewness changes, but not mean MT changes, were spatially associated with oligodendrocyte gene expression patterns. Additional analyses indicated that skewness effects occurred rather independently of MRI-based measures of cortical interface blurring and variations in cortical thickness, supporting specificity to intracortical microstructure. Our findings may align with several mechanisms of experience-dependent myelination (*de Faria et al., 2018*). Increasing mean MT likely reflects myelination of previously unmyelinated axons. On the other hand, as skewness changes persisted even after controlling for changes in the mean, these more likely capture architectural reconfigurations or laminar specific changes in axonal myelination. Underlying mechanisms may include de novo generation of oligodendrocytes (*Zuccaro and Arlotta, 2013*), activity-dependent modifications in neuronal and oligodendrocyte precursors (*Gibson et al., 2014*), and tuning of the distance between nodes of Ranvier along both cortico-cortical as well as long-range axons (*Arancibia-Cárcamo et al., 2017*). The putative functional role of myelin in cortical grey matter may be to insulate fibres from making new synaptic connections, thus enhancing stability (*Braitenberg, 1962*; *Braitenberg and Schüz, 1998*; *Micheva et al., 2016*). Conversely, overall lower myelin content in transmodal regions might indicate a higher potential for plasticity (*Huntenburg et al., 2017*), possibly making these areas more suitable to support adaptive behaviour and learning that are core to many integrative functions that mature during adolescence. Rodent experiments suggest that experience-induced oligodendrocyte remodelling supports the development of positive social behaviour (*Makinodan et al., 2012*).

Conversely, abnormal myelination has been implicated in several brain disorders, however, studies on intracortical myeloarchitecture are scarce (*Mighdoll et al., 2015*; *Stedehouder and Kushner, 2017*; *Bernhardt et al., 2018*). The architectural changes observed in the present cohort may represent a microstructural substrate that could determine overall cognitive and social capacities or susceptibility to a range of neuropsychiatric and neurological conditions occurring in late childhood and adolescence.

Adaptation of a recent microstructure profile covariance analysis framework to longitudinal MT data illustrated the macroscale impact of microstructural changes. Specifically, decreases in MT profile skewness pushed regions to a more sensory-like architecture (which was coupled with high mean MT). In contrast, increases in skewness pushed regions towards a paralimbic-like MT profile. These regions also underwent protracted increases in mean MT, but still did not reach the high levels observed in sensory regions. These two regionally distinct developmental patterns within the association cortex promoted a more bimodal distribution of myeloarchitectural types across the cortex. Similar properties of modular segregation in tractography-based networks have been shown to mediate age-related improvements in executive function throughout adolescence (*Baum et al., 2017*), reinforcing the importance of these processes for understanding psychosocial maturation. Drifts in association cortex towards either sensory-like or paralimbic-like architecture represents, thus, an expansion of the sensory-fugal gradient of microstructural differentiation (*Paquola et al., 2019*). Such a sensory-fugal gradient was previously described by *Mesulam (1998)* based on non-human primate research to encapsulate cortex-wide variations in architecture and connectivity, and has since been suggested to reflect increasing synaptic plasticity towards transmodal regions (*García-Cabezas et al., 2017*). Systematic variations in the degree of experience-dependent plasticity may explain why myelin-derived markers develop differently along the sensory-fugal gradient (*de Faria et al., 2018*), in contrast to other known developmental gradients such as the rostro-caudal timing of terminal neurogenesis (*Rakic, 1974*). Differentiation of association cortices also conforms with notions of the 'tethering hypothesis' of cortical evolution (*Buckner and Krienen, 2013*). According to this framework, intermediate regions of the cortical hierarchy are less constrained by extrinsic inputs and intrinsic signaling molecules, which has allowed for massive surface area expansion throughout mammalian evolution. Our findings suggest that reduced constraints on association cortices allow for protracted development of myeloarchitecture, whereas the sensory and limbic anchors are well-defined prior to adolescence.

Our work suggests that myelination during adolescence is unlikely to be a question of simply more or less. Instead, our longitudinal findings show that the type of change is topologically divergent when we take depth into consideration. Expanding upon previous evidence of adolescent increases in overall mean intracortical myelin content, our findings demonstrate a relative preferential specific accumulation of myelin towards mid-to-deeper infragranular layers mainly in association cortices. As our analyses have shown, these findings are not explainable by changes in overall cortical morphology during adolescence, but instead likely reflect architectural changes associated with oligodendrocyte related processes. The coordinated change patterns strengthen the notion that the forces of adolescent development further widen the existing axis of macroscale cortical organization, driving association cortex either closer towards sensory or limbic systems. Thus, our findings illustrate how cell-type and layer specific microstructural changes assessed in the direction of cortical columns contribute to the maturation of macroscale cortical organisation and suggest adolescent calibration of structural hierarchical gradients.

## Materials and methods

### Imaging acquisition and processing

The present study included a subset of individuals from the NeuroScience in Psychiatry Network (NSPN) study (*Whitaker et al., 2016*; *Kiddle et al., 2018*). For a visualization of the sampling design, see *Appendix 1—figure 6*. In brief, the NSPN study comprises a primary cohort of 2402 healthy young people, recruited from schools, colleges, NHS primary care services and direct advertisements in north London and Cambridgeshire. Participants were stratified into five age groups (14–15, 16–17, 18–19, 20–21 and 22–25 years) and each stratum was evenly balanced for sex and ethnicity. Primary participants completed demographic, medical, childhood trauma and mental health

questionnaires by post. The secondary cohort sub-sampled approximately 60 individuals from each stratum in the primary cohort, maintaining the sex and ethnicity balance. Secondary participants completed MRI scanning as part of a whole-day assessment at one of two sites (Cambridge and London, UK), on at least two occasions. Cohort retention for the MRI follow-up was 74% (*Kiddle et al., 2018*). Further inclusion criteria for the present study were availability of T1w and MT data available at two timepoints, resulting in 234 healthy adolescents. Following additional quality control on raw images, surface reconstructions and MT profiles, detailed below, the sample included 180 individuals (age stratification at baseline, n = 39/41/39/33/26, inter-scan interval = 15.4 ± 3.5 months, 86 females).

T1w and MT imaging were acquired as part of the quantitative multiparameter mapping (MPM) sequence (*Weiskopf et al., 2013*) on three identical TIM Trio 3T scanners; two located in Cambridge and one located in London. The MPM sequence comprises several multi-echo 3D FLASH (fast low angle shot) scans (*Weiskopf et al., 2013*). MT-weighting was achieved by applying an off-resonance Gaussian-shaped RF pulse (4 ms duration, 220° nominal flip angle, 2 kHz frequency offset from water resonance) prior to the excitation with TR/α = 23.7 ms/6°. Multiple gradient echoes were acquired with alternating readout polarity at six equidistant echo times (TE) between 2.2 and 14.7 ms. for MT weighted acquisition. The MT saturation parameter decouples the MT signal from the longitudinal relaxation rate, making it a semi-quantitative measure that is robust to relaxation times and field inhomogeneities (*Weiskopf et al., 2013*; *Hagiwara et al., 2018*). Other acquisition parameters were: 1 mm isotropic resolution, 176 sagittal partitions, field of view (FOV) = 256 × 240 mm, matrix = 256 × 240×176, parallel imaging using GRAPPA factor two in phase-encoding (PE) direction (AP), 6/8 partial Fourier in partition direction, non-selective RF excitation, readout bandwidth BW = 425 Hz/pixel, RF spoiling phase increment = 50°.

All raw images were visually inspected by experienced researchers and 11 participants were excluded due to excessive motion artefacts. Surface reconstructions were visually inspected and manually edited for all scans, up to 10 times (*Romero-Garcia et al., 2018*). At each iteration, control points and grey/white matter edits were included, and the surface reconstruction was repeated. 10 participants were excluded due to poor surface reconstructions. Following cortical surface reconstruction and surface-based co-registration between T1w and MT weighted scans, we generated 14 equivolumetric cortical surfaces within the cortex (*Wagstyl et al., 2018*), and systematically sampled MT intensity along these surfaces (*Paquola et al., 2019*; *Figure 1*). Next, depth-wise MT profiles were calculated across all vertices in native space and MT profiles were averaged within 1012 equally sized nodes (*Figure 1*). Given 1 mm isotropic voxels and 1.4–4.2 mm cortical thickness in present dataset, we estimate that each vertex-wise depth profile contains approximately 2–5 voxels. Using trilinear volume interpolation to each intracortical surface allowed for greater nuance in the intensity estimates along the depth profile. In total, each parcel-wise profile contains 71 ± 5.4 voxels. The parcellation scheme was mapped from a standard space (fsaverage) to native space for each subject using surface-based registration (*Hong et al., 2017*). We corrected for depth-specific partial volume effects (PVE) of cerebrospinal fluid using a mixed tissue class model (*Kim et al., 2005*) to reduce potential bias of averaging MT values in voxels with CSF (*Mossahebi et al., 2015*). Specifically, we fitted a linear model at each node (n) and each surface (s) of the form

$$\mathrm{MT(n,s)} \sim \mathrm{b_0} + \mathrm{b_1 CSF(n,s)}$$

where MT(n,s) and CSF(n,s) represent node-specific, surface-specific MT value and CSF partial volume effect estimates. Final CSF-corrected MT values were calculated as the sum of the residuals [MTc(n,s)=T1(n,s) – (b0+b1*CSF(n,s))] and the uncorrected group average MT value.

## Baseline analysis of MT profiles

We characterised the MT profiles by the central moments of the intensity distributions; mean, standard deviation (SD), skewness, and kurtosis (*Zilles et al., 2002*). We focused our main analysis on the first and third moments (*Figure 1*), owing to the collinearity of SD with mean and kurtosis with skewness (see also *Appendix 1—figure 1*), the clearer biological interpretation of mean and skewness for the MT profiles, as well as how the first and third moments capture different dimensions of the intensity distribution. Prior to statistical assessment, we identified and removed outlier individuals (n = 33). Outliers were defined as individuals with >1% nodes that deviated from the age-

stratified median of nodal mean MT by >3 interquartile ranges. In an effort to validate the relationship between MT and myelin, which is discussed in more detail elsewhere (*Heath et al., 2018*; *Schmierer et al., 2007*; *Weiskopf et al., 2013*), we examined MT-derived and post mortem derived myelin profiles in matched regions of the cortex. To further understand architectural and cellular underpinnings of the in vivo MT profiles, we assessed the similarity of the MT-derived profiles with intracortical myelin profiles computed from previously published post mortem myelin sections. While there are clear differences in the resolution and specificity to myelin between the histological staining and our in vivo proxy data, findings were overall supportive of a close association between in vivo MT profiles and those derived from myelin stained sections. This line of evidence supports the specificity of the MT features to intracortical myeloarchitecture and motivates efforts to increase the availability of post mortem histological measures for more detailed cross-validation studies of putative MRI markers of myelin. We selected hyper-stained myelin pictures from classical literature (*Vogt and Vogt, 1919*; *Vogt, 1911*) that represent different levels of cytoarchitectural complexity and have been well characterised in recent work (*Palomero-Gallagher and Zilles, 2019*). We extracted a rectangle section of each image (spanning from the pial layer to the white matter boundary; *Figure 1A*), inverted the colour to align with MT imaging (*i.e.,:* myelin is more white), obtained intensity values per pixel by reading the image into MATLAB (v2017b), then generated a region-specific intensity profile by averaging values row-wise. Finally, we calculated the MT moments for each region and contrasted these values with baseline group-average MT moments for matched regions. Additionally, we contrasted baseline group-average MT moments across levels of laminar differentiation (*Mesulam, 2002*; *Paquola et al., 2019*) and cytoarchitectonic type (*Von Economo and Koskinas, 2008*; *Whitaker et al., 2016*) to determine whether the in vivo derived MT moments recapitulate histological evidence of cytoarchitectonic variation. MPC networks were calculated as the pairwise Pearson correlation between nodal MT profiles, controlling for the average whole-cortex MT profile. In line with previous work (*Paquola et al., 2019*), we performed nonlinear dimensionality reduction to characterise the principle gradient of variance in MPC. We assessed the correspondence of the MPC gradient with the MT moments using a node-wise Spearman correlations, with p-values determined against 10000 null models obtained from spin permutations (*Alexander-Bloch et al., 2018*). All spatial correlations were estimated and tested in the same way throughout the study.

## Longitudinal assessment of age-related changes in MT moments

Age-related changes in MT moments were estimated at each node within four linear mixed effect models (LME) using SurfStat (*Worsley et al., 2009*) for Matlab, accounting for the non-independence of subjects as well as sex, using the following model:

$$\text{moment(n)} \sim b_0 + b_1 \text{age} + b_2 \text{sex} + (1|\text{subject} + \varepsilon)$$

where *n* represents the node. The addition of person-specific random intercepts significantly improves model fit in accelerated longitudinal designs. LMEs were performed on each node and t-statistics were projected onto the cortical surface. Additionally, we examined the relationship between level of laminar differentiation and adolescent development of MT moments. To assess the relative over- or under-representation of significant regions per laminar class we generated 10,000 spin permutations of the *t*-maps, which controls for spatial contiguity and hemispheric symmetry across permutations (*Alexander-Bloch et al., 2018*; *Seidlitz et al., 2019*; *Váša et al., 2018*). Specifically, we differently rotated the *t*-maps 10,000 times and for each permutation of this spin or rotation we computed counts per laminar class that passed a false discovery rate (FDR) correction for multiple comparisons (*Benjamini and Hochberg, 1995*) to create a null-distribution of the counts table. Based on the null distribution, we computed Z-scores and two-sided *p*-values for the actual count table and corrected for multiple comparisons across the entire table using conservative Bonferroni correction.

## Gene expression analyses

We measured the spatial overlap between our baseline maps of MT moments (e.g., one scan per subject; *Figure 1E*) and t-statistics (*Figure 2A*) and maps of post mortem gene expression from the Allen Institute for Brain Sciences (AIBS). We used Neurovault gene decoding of the AIBS dataset to

identify the significant associations of spatial gene expression with t-maps (*Hawrylycz et al., 2012*; *Gorgolewski et al., 2015*). Neurovault implements a non-linear co-registration of each AIBS donor brain to a standard MNI template. Visual inspection revealed that one donor required additional manual correction due to a cerebellar deformations. Then, 4 mm spheres were drawn around each probe coordinate and values from the input t-maps, already in MNI space, were averaged within each sphere. Mixed effect analysis modelled the associations between each individual gene across the six donors and each input t-map. Meta-data from the gene probes (i.e., their EntrezID's) were downloaded within the same pipeline and thus always had the latest information available from AIBS. Genes without EntrezID were excluded from subsequent analysis. Filtering was omitted on the genes to ensure a fully data-driven approach. Resultant gene associations p-values were corrected at an FDR level of 0.05 across all included genes and only genes passing correction were included in subsequent analyses. Enrichment analyses were conducted using Enrichr (*Chen et al., 2013*; *Kuleshov et al., 2016*), using a Z-score modification of Fisher's exact test and FDR correction. It should be noted that the AIBS data are based on adult post-mortem and that the developmental associations are indirect associations between the enriched gene-set from spatial overlap analysis and a different developmental gene-expression dataset. Specifically, the t-statistic maps of age-related changes in the moments were spatially correlated with gene expression maps from the AIBS dataset using Neurovault (https://github.com/NeuroVault/NeuroVault/tree/master/ahba_docker). The genes derived from this spatial analysis were subsequently compared against developmental expression profiles from the BrainSpan dataset (http://www.brainspan.org/) using the CSEA developmental expression tool developed by the Dougherty lab (http://genetics.wustl.edu/jdlab/csea-tool-2/).

## Spatial topography of age-related changes in MT profiles

Age-related changes in inter-regional microstructural similarity were assessed by applying the same LME to individual edges of subject specific MPC networks (*Paquola et al., 2019*). Age-related changes were deemed significant using a two-tailed $q_{FDR} < 0.05$. Diffusion map embedding, a non-linear dimensionality reduction technique (*Coifman and Lafon, 2006*), was applied to the resultant t-statistic matrix to discern the spatial topography of age-related changes. The first principal component represents the principal axis of variation in age-related changes in MPC. Nodal loadings onto the principal component, otherwise known as gradient values ($G1_{DEV}$), depict the similarity of nodal patterns of MPC change. Similar to previous analyses (*Paquola et al., 2019*), we examined whether $G1_{DEV}$ differed across levels of laminar differentiation using ridge plots (*Wilke, 2018*). For closer inspection of MT profile changes across the developmental gradient, we generated group-average MT profiles for the youngest (<16 years, n = 43) and oldest (>25 years, n = 30) age strata. To probe the impact of developmental differentiation on the maturity of the underlying microstructure map, we generated group-average MPC matrices within the youngest, oldest as well as a mid-range (20–22 years) age-strata. MPC matrices were subjected to diffusion map embedding (*Vos de Wael et al., 2019*), then the young and old embeddings were aligned to the mid-range embedding using Procrustes rotation (*Langs et al., 2015*). We rank ordered and binned G1into ten, approximately equal sized bins (nodes per bin ~110). Age-related differences in MPC gradients were assessed by (i) bin-wise difference average $G1_{MPC}$ values, (ii) bin-wise Cohen's d effect size change of gradient values, (iii) node-wise difference in MPC to the gradient anchors, that is the extreme bins, and (iv) bin-wise difference in MT moment values. To reconcile macroscale changes in the MPC gradients to microstructural alterations, we performed a spatial Pearson correlation between unthresholded t-statistic maps of age-related changes in MT mean and skewness (from *Figure 2B*) with shifts relative to anchors (map iii, *Figure 4B* left).

## Data and code availability

Preprocessed microstructure profiles are available via GitHub: https://github.com/MICA-MNI/micaopen/tree/master/a_moment_of_change (*Paquola, 2019*; copy archived at https://github.com/elifesciences-publications/micaopen/tree/master/a_moment_of_change).

The repository also includes Matlab and R scripts to reproduce the primary analyses and figures.

## Acknowledgements

CP is supported by the Fonds de la Recherche du Quebec – Santé (FRQS), RAIB is supported by a British Academy Postdoctoral fellowship and Autism Research Trust. CP, RAIB, GW, BB are supported by a Cambridge-MNI collaborative research grant. RRG was funded by the NSPN and the Guarantors of Brain. JS was supported by the NIH Oxford-Cambridge Scholars' Program. KW was supported by the Health Brain Healthy Lives (HBHL) Initiative. KJW was funded by a The Alan Turing Institute under the EPSRC grant EP/N510129/1. PEV was supported by the MRC (MR/K020706/1), is a Fellow of MQ: Transforming Mental Health (MQF17/24) and is a Fellow of the Alan Turing Institute funded under EPSRC (EP/N510129/1). BB is supported by National Science and Engineering Research Council of Canada (NSERC, Discovery-1304413), the Canadian Institutes of Health Research (CIHR, FDN-154298), the Azrieli Center for Autism Research of the Montreal Neurological Institute (ACAR), SickKids Foundation (NI17-039), and received salary support from FRQS (Chercheur Boursier Junior 1). EB is supported by a NIHR Senior Investigator award.

Data were curated and analysed using a computational facility funded by an MRC research infrastructure award (MR/M009041/1) and supported by the NIHR Cambridge BRC. This work was supported by the Neuroscience in Psychiatry Network (NSPN) Consortium, a strategic award from the Wellcome Trust to the University of Cambridge and University College London (095844/Z/11/Z); by the Cambridge NIHR Biomedical Research Centre. The views expressed are those of the authors and not necessarily those of the NHS, the NIHR or the Department of Health and Social Care.

## Additional information

### Funding

| Funder | Grant reference number | Author |
| --- | --- | --- |
| Wellcome | 095844/Z/11/Z | Rafael Romero-Garcia<br>Edward T Bullmore |
| Fonds de la recherche en sante du Quebec | | Casey Paquola |
| Medical Research Council | MR/K020706/1 | Petra E Vértes |
| British Academy | | Richard AI Bethlehem |
| Autism Research Trust | | Richard AI Bethlehem |
| Cambridge-Montreal Neurological Institute and Hospital | | Casey Paquola<br>Richard AI Bethlehem<br>Guy B Williams<br>Boris Bernhardt |
| Guarantors of Brain | | Rafael Romero-Garcia |
| National Institutes of Health | Oxford-Cambridge Scholars' Program | Jakob Seidlitz |
| Healthy Brains, Healthy Lives | | Konrad Wagstyl |
| Engineering and Physical Sciences Research Council | EP/N510129/1 | Kirstie J Whitaker<br>Petra E Vértes |
| MQ: Transforming Mental Health | MQF17/24 | Petra E Vértes |
| Natural Sciences and Engineering Research Council of Canada | Discovery-1304413 | Boris Bernhardt |
| Canadian Institutes of Health Research | FDN-154298 | Boris Bernhardt |
| Azrieli Center for Autism Research of the Montreal Neurological Institute | | Boris Bernhardt |
| Sick Kids Foundation | NI17-039 | Boris Bernhardt |

| Fonds de Recherche du Qué-bec - Santé | Chercheur Boursier Junior 1 | Boris Bernhardt |
| National Institute for Health Research | Senior Investigator Award | Edward T Bullmore |
| Cambridge NIHR Biomedical Research Centre | | Edward T Bullmore |

The funders had no role in study design, data collection and interpretation, or the decision to submit the work for publication.

## Author contributions

Casey Paquola, Conceptualization, Data curation, Formal analysis, Investigation, Visualization, Methodology, Writing—original draft, Writing—review and editing; Richard AI Bethlehem, Conceptualization, Formal analysis, Investigation, Visualization, Writing—original draft, Writing—review and editing; Jakob Seidlitz, Formal analysis, Methodology, Writing—review and editing; Konrad Wagstyl, Investigation, Methodology, Writing—review and editing; Rafael Romero-Garcia, Daniel S Margulies, Methodology, Writing—review and editing; Kirstie J Whitaker, Petra E Vértes, Data curation, Writing—review and editing; Reinder Vos de Wael, Software, Methodology; Guy B Williams, Funding acquisition, Writing—review and editing; NSPN Consortium, Funding acquisition; Data curation; Boris Bernhardt, Conceptualization, Resources, Supervision, Funding acquisition, Investigation, Visualization, Methodology, Writing—review and editing; Edward T Bullmore, Conceptualization, Resources, Data curation, Supervision, Funding acquisition, Investigation, Writing—review and editing

## Author ORCIDs

Casey Paquola https://orcid.org/0000-0002-0190-4103
Richard AI Bethlehem https://orcid.org/0000-0002-0714-0685
Jakob Seidlitz https://orcid.org/0000-0002-8164-7476
Petra E Vértes https://orcid.org/0000-0002-0992-3210
Boris Bernhardt https://orcid.org/0000-0001-9256-6041

## Ethics

Human subjects: Participants provided informed written consent for each aspect of the study, and parental consent was obtained for those aged 14-15 y old. Ethical approval was granted for this study by the NHS NRES Committee East of England-Cambridge Central (project ID 97546). The authors assert that all procedures contributing to this work comply with the ethical standards of the relevant national and institutional committees on human experimentation and with the Helsinki Declaration of 1975, as revised in 2008.

## Decision letter and Author response

Decision letter https://doi.org/10.7554/eLife.50482.023
Author response https://doi.org/10.7554/eLife.50482.024

# Additional files

## Supplementary files

• Supplementary file 1. Supplementary materials.
DOI: https://doi.org/10.7554/eLife.50482.013

• Transparent reporting form DOI: https://doi.org/10.7554/eLife.50482.014

## Data availability

Preprocessed microstructure profiles are available via GitHub: https://github.com/MICA-MNI/micaopen/tree/master/a_moment_of_change (copy archived at https://github.com/elifesciences-publications/micaopen/tree/master/a_moment_of_change). The repository also includes Matlab and R scripts to reproduce the primary analyses and figures.

The following datasets were generated:

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

## Appendix 1

DOI: https://doi.org/10.7554/eLife.50482.015

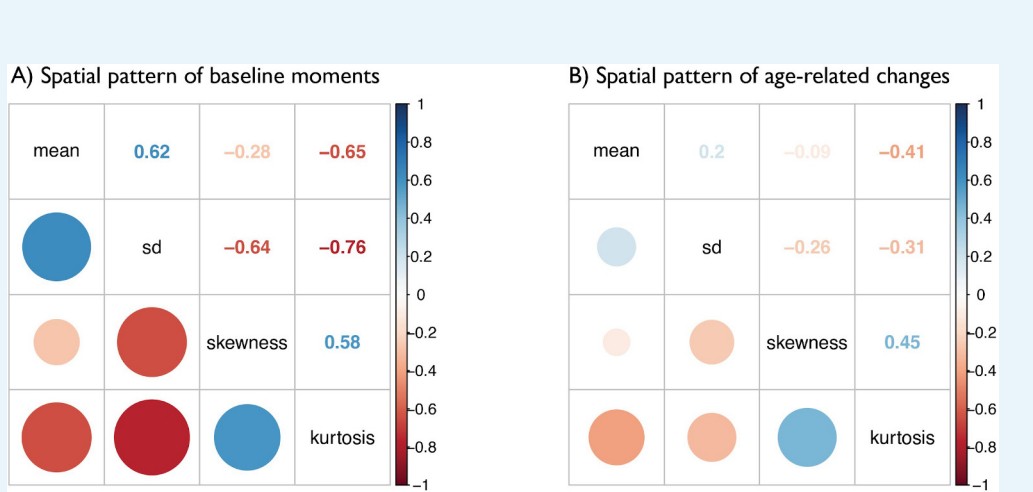

**Appendix 1—Figure 1.** Matrices depict the strength of correlations (r values) between (**A**) baseline maps of MT moments and (**B**) t-statistic maps of age-related changes in MT moments.

DOI: https://doi.org/10.7554/eLife.50482.016

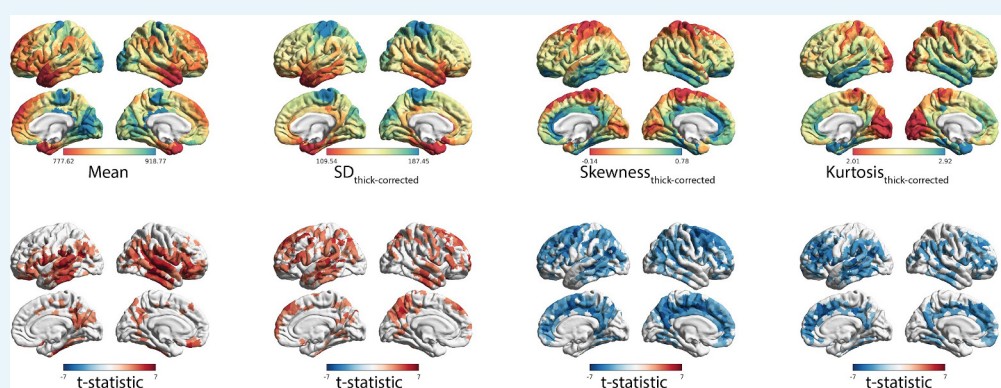

**Appendix 1—Figure 2.** Independence of effects controlling for cortical thickness. (**A**) Spatial distribution of MT moments corrected for cortical thickness ($moment(n){\sim}b_0 + b_1 thickness(n) + \varepsilon$). (**B**) t-statistics representing the age-related changes in MT moments, with cortical thickness included in the linear mixed effect model as a covariate ($moment(n) \sim b_0 + b_1 thickness(n) + b_2 age + b_3 sex + (1|subject) + \varepsilon$).

DOI: https://doi.org/10.7554/eLife.50482.017

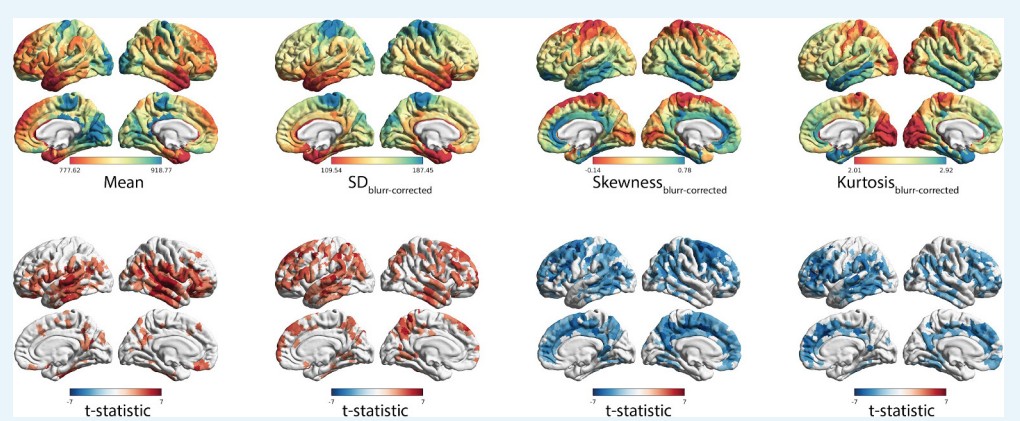

**Appendix 1—Figure 3.** Independence of effects controlling for interface blurring. (**A**) Spatial distribution of MT moments corrected for interface blurring ($moment(n){\sim}b_0 + blurring(n) + \varepsilon$). (**B**) $t$-statistics representing the age-related changes in MT moments, with interface blurring included in the linear mixed effect model as a covariate ($moment(n) \sim b_0 + b_1 blurring(n) + b_2 age + b_3 sex + (1|subject) + \varepsilon$).

DOI: https://doi.org/10.7554/eLife.50482.018

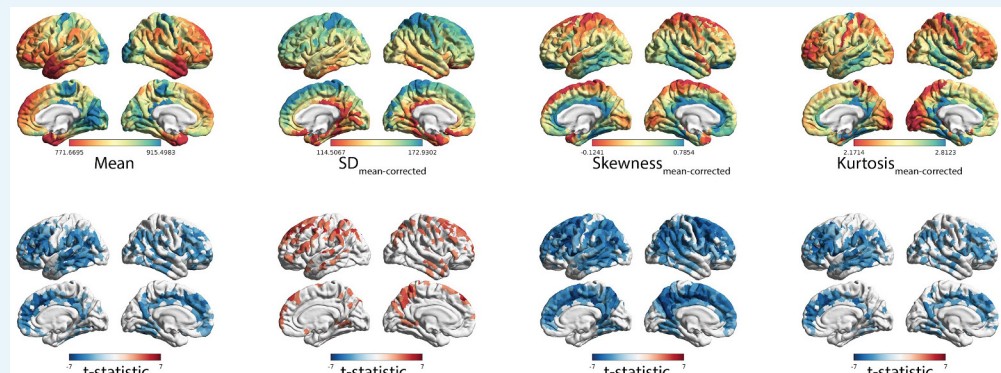

**Appendix 1—Figure 4.** Independence of effects controlling for mean MT. (**A**) Spatial distribution of MT moments corrected for mean MT ($moment(n){\sim}b_0 + b_1 meanMT(n) + \varepsilon$). (**B**) $t$-statistics representing the age-related changes in MT moments, with mean MT included in the linear mixed effect model as a covariate ($moment(n) \sim b_0 + b_1 meanMT(n) + b_2 age + b_3 sex + (1|subject) + \varepsilon$).

DOI: https://doi.org/10.7554/eLife.50482.019

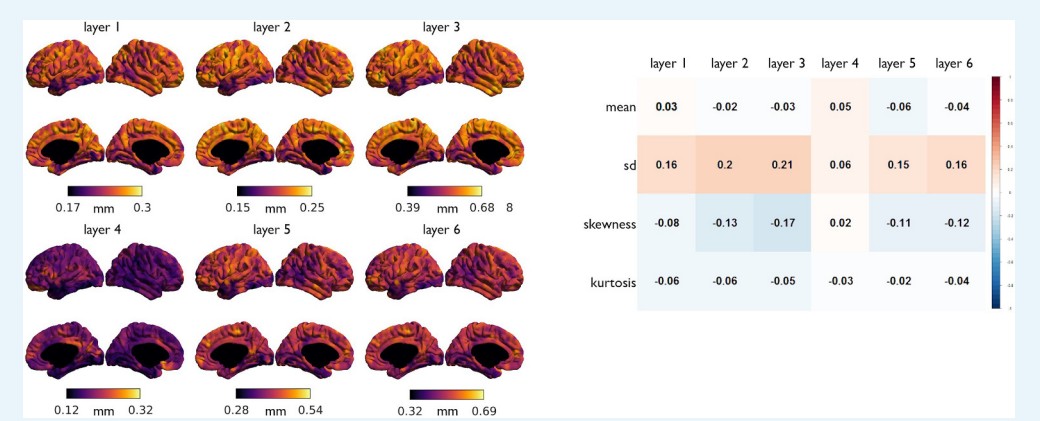

**Appendix 1—Figure 5.** Association of laminar thickness with MT moments. (*Left*) Laminar thickness, derived from a post mortem volumetric reconstruction of a Merker stained human brain (*Wagstyl et al., 2018*). (*Right*) Spearman correlation coefficients between spatial maps of baseline MT moments with laminar thicknesses.
DOI: https://doi.org/10.7554/eLife.50482.020

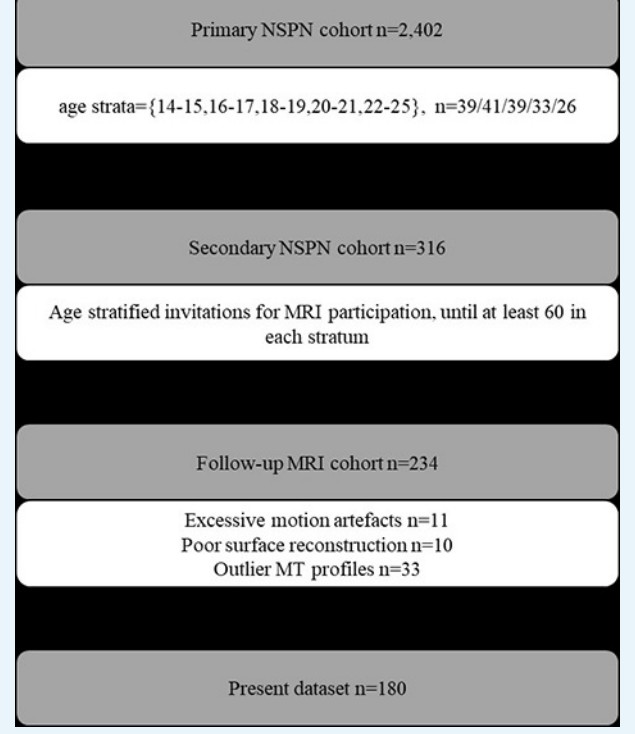

**Appendix 1—Figure 6.** Sampling design.
DOI: https://doi.org/10.7554/eLife.50482.021

