## [Decision Letter]

**Acceptance Summary:**

This novel and important study capitalizes upon a large sample of youths scanned twice with magnetization transfer imaging, in order to map developmental profiles of cortical myelin. The authors use this approach to demonstrate that the human cortex undergoes dissociable, depth-specific changes in cortical myeloarchitecture during development. Critically, greater imaging markers of myelin were spatially associated with gene expression markers of oligodentrocytes, and align with a gradient ranging from primary sensory to higher-order association cortex. Together, this approach provides a new understanding of how cortical myelin develops during the critical period of adolescence. Furthermore, this data has numerous implications for understanding both healthy brain maturation and abnormal brain development associated with neuropsychiatric syndromes.

**Decision letter after peer review:**

Thank you for submitting your article "Shifts in myeloarchitecture characterise adolescent development of cortical gradients" for consideration by *eLife*. Your article has been reviewed by Joshua Gold as the Senior Editor, a Reviewing Editor, and two reviewers. The following individuals involved in review of your submission have agreed to reveal their identity: Kate Mills (Reviewer #1); Ben Fulcher (Reviewer #2).

The reviewers have discussed the reviews with one another and the Reviewing Editor has drafted this decision to help you prepare a revised submission.

Summary:

This is a study investigating intracortical microstructure in a sample of 223 individuals scanned twice between ages 14-27 years. Authors estimated myelin content in the cortex of magnetic resonance images using magnetization transfer, and by using central moments to test the hypothesis that mean and skewness of the magnetization transfer profiles would show different developmental trajectories and relate to different gene expression patterns. The study is comprehensive and the methods are sound. Writing and presentation are excellent, and code for reproducing the analyses is made available (and processed data where possible), further enhancing the impact of this work. Nonetheless, reviewers felt that several aspects of the work could be strengthened on revision.

Essential revisions:

1) NSPN sample. More details about the full sample (the full NSPN sample) compared to the included sample for this study would be useful. A visualization of the sampling design would be helpful.

2) Quality control. More details regarding the exclusion criteria and brain image quality control procedures are needed. What was the procedure for inspection, criteria used to determine exclusion, and number of scans excluded? Were both the raw images and processed images inspected for quality? Please describe the extent of manual intervention of processed images, including the protocol and number of scans that were successfully processed post-intervention and included in analyses.

3) AIBS. Further details regarding the processing of the gene expression maps is necessary (see: Arnatkeviciūtė et al., 2019). Furthermore, authors should clearly state that AIBS data are based on adult postmortem oligodendrocyte density. As such, it seems warranted to temper their claims (example: "confirmed a spatio-temporal overlap of our findings from NSPN with *myelin processes during adolescence*").

4) MT and myelin. The authors should clarify how sensitive is MT to myelin. The strength of this link is crucial to a lot of the interpretation and biological novelty of the results. MT (like T1w:T2w) may be sensitive not just to myelin, but also potentially other biological variables. Some evidence/discussion of this, and the extent to which the reader can safely make the association to myelin content would be helpful in the introduction. Furthermore, many features are different between the myelin stains and MT profiles (Figure 1B). Some brief discussion of the differences would be helpful. Finally, it would be useful to clarify if to the author's knowledge this is the first time anyone has matched MT depth profiles to a histological measurement (or discuss relevant literature of past efforts).

5) MT skewness and relative layer thickness. MT skewness may depend on the relative thickness of the different layers. It would be useful to clarify how much results explain the data beyond a simple cytoarchitectural analysis (e.g., are the depth profiles acting as a quantitative substitute for cytoarchitecture 'types', or is the inferred connection to myeloarchitecture relevant?) Furthermore, it would be useful to clarify if every node is squashed to the same absolute depth? This could impact interpretation: two areas with an identical ratio of myelination in deep relative to superficial layers would have different skewnesses if their middle layer was made thicker or thinner (thicker middle layer could push out the skewness). If this is the case, MT skewness could not simply be interpreted as a "ratio of myelination in deeper compared to more superficial layers"; it would more represent a measure of the preferential distribution of MT closer to the white matter boundary. Clarification is needed.

6) Statistical testing. Every time a statistical analysis is performed (and a test statistic/p-value quoted), it should be clearly stated to the reader what it is. Non-normal distributions seem common in this data; Spearman's correlations may be appropriate. Furthermore, it appears that the spin-test based permutations are only used for some analyses but not others (e.g., only from the "Age-related changes… MT profiles" section onwards). Were spin tests used for earlier spatial association analyses? If not, can this be justified?

7) Utility of low-dimensional embedding. Regarding this statement: "Nodes closer in this embedding space increase in microstructural similarity during adolescence, whereas distant nodes decouple", it is unclear why a low-dimensional embedding is useful a direct measurement of change is available. Couldn't you more easily do an analysis directly (e.g., investigate the structure of the 'increasing' edges?) What is gained from this harder-to-interpret nonlinear embedding that helps test/interpret your hypotheses? Perhaps more importantly, it is unclear if this statement is supported by the data: Wouldn't the embedding put nodes close in the space that have similar "patterns" of age-related MPC change with other network nodes? In the plot of the behavior of G1_dev, it does not display the claimed behavior (Figure 4B) – instead, it seems like the largest values of G1_dev already have ~maximal G1_MPC, and thus barely change at all between the <16 and >24 groups. Clarification is warranted.

8) Biological interpretation. In the discussion, the biological interpretation of the results at times is rather thin. If MT is a strong myelin marker, then why would myeloarchitecture be reorganizing like this (e.g., to support what behavior)? Axonal myelination improves transmission speeds and prevents the formation of new synaptic connections. Could you speculate on how this might alter behavior relevant to the longitudinal changes analyzed? Relevant papers to consider include:

*Micheva et al., (2016).

*Braitenberg, (1962).

*Huntenburg et al., (2017).

---

## [Author Response]

Summary:This is a study investigating intracortical microstructure in a sample of 223 individuals scanned twice between ages 14-27 years. Authors estimated myelin content in the cortex of magnetic resonance images using magnetization transfer, and by using central moments to test the hypothesis that mean and skewness of the magnetization transfer profiles would show different developmental trajectories and relate to different gene expression patterns. The study is comprehensive and the methods are sound. Writing and presentation are excellent, and code for reproducing the analyses is made available (and processed data where possible), further enhancing the impact of this work. Nonetheless, reviewers felt that several aspects of the work could be strengthened on revision.

We thank the reviewers and Editor for recognising the quality and impact of our work. We are grateful to the reviewers for their constructive comments, which we feel have improved the manuscript. We have addressed all comments point-by-point.

Essential revisions:1) NSPN sample. More details about the full sample (the full NSPN sample) compared to the included sample for this study would be useful. A visualization of the sampling design would be helpful.

We thank the reviewers for this suggestion and have provided greater characterisation of our cohort in the Materials and methods section. A visualization is provided in the new Appendix—figure 6.

“The present study included a subset of individuals from the NeuroScience in Psychiatry Network (NSPN) study (Whitaker et al., 2016; Kiddle et al., 2018). […] Secondary cohort data were eligible for inclusion in this study if T1w and MT data were available at two timepoints, resulting in N=234 healthy adolescents. Following additional quality control on raw images, surface reconstructions and MT profiles, detailed below, the evaluable sample included N=180 individuals (age stratification at baseline, n=39/41/39/33/26, inter-scan interval=15.4 ± 3.5 months, 86 females).”

2) Quality control. More details regarding the exclusion criteria and brain image quality control procedures are needed. What was the procedure for inspection, criteria used to determine exclusion, and number of scans excluded? Were both the raw images and processed images inspected for quality? Please describe the extent of manual intervention of processed images, including the protocol and number of scans that were successfully processed post-intervention and included in analyses.

To clarify, quality control and exclusion criteria were conducted at three stages: (a) on the raw images, (b) at the level surface reconstructions, and c) with respect to the MT profiles. As suggested, we have now provided more specifics in the revised Materials and methods section:

“All raw images were visually inspected by experienced researchers and 11 participants were excluded due to excessive motion artefacts. Surface reconstructions were visually inspected and manually edited for all scans, up to 10 times (Romero-Garcia et al., 2018). At each iteration, control points and grey/white matter edits were included, and the surface reconstruction was repeated. 10 participants were excluded due to poor surface reconstructions.”

“Prior to statistical assessment, we identified and removed outlier individuals (n=33). Outliers were defined as individuals with >1% nodes that deviated from the age-stratified median of nodal mean MT by >3 interquartile ranges.”

3) AIBS. Further details regarding the processing of the gene expression maps is necessary (see: Arnatkeviciūtė et al., 2019). Furthermore, authors should clearly state that AIBS data are based on adult postmortem oligodendrocyte density. As such, it seems warranted to temper their claims (example: "confirmed a spatio-temporal overlap of our findings from NSPN with myelin processes during adolescence").

We thank the reviewers for this important comment and are happy to provide further clarifications. The present analysis relied on the Neurovault implementation of AIBS gene expression processing. More details regarding the automated gene decoding are provided in the Materials and methods section:

“We measured the spatial overlap between our baseline maps of MT moments (e.g., one scan per subject; Figure 1E) and t-statistics (Figure 2A) and maps of post mortem gene expression from the Allen Institute for Brain Sciences (AIBS). […] Resultant gene associations p-values were corrected at an FDR level of 0.05 across included genes and only genes passing correction were included.”

As suggested, we tempered claims regarding the adult post-mortem nature of gene expression data in the updated Results section:

“Both analyses thus confirmed a spatial overlap of our findings from NSPN with genes associated with myelin processes that are also enriched during adolescence. […] While there was thus evidence for an adolescent developmental signal, results are nevertheless indirect also given that glial and oligodendrocyte associated genes may undergo expression changes during other periods of the lifespan (Soreq et al., 2017).”

We also added the following caveat to the Materials and methods section:

“It should be noted that the AIBS data is based on adult post-mortem and that the developmental associations are indirect associations between the enriched gene-set from spatial overlap analysis and a different developmental gene-expression dataset. Specifically, the t-statistic maps of age-related changes in the moments were spatially correlated with gene expression maps from the AIBS dataset using Neurovault (https://github.com/NeuroVault/NeuroVault/tree/master/ahba_docker). The genes derived from this spatial analysis were subsequently compared against developmental expression profiles from the BrainSpan dataset (http://www.brainspan.org/) using the CSEA developmental expression tool developed by the Dougherty lab (http://genetics.wustl.edu/jdlab/csea-tool-2/).”

4) MT and myelin. The authors should clarify how sensitive is MT to myelin. The strength of this link is crucial to a lot of the interpretation and biological novelty of the results. MT (like T1w:T2w) may be sensitive not just to myelin, but also potentially other biological variables. Some evidence/discussion of this, and the extent to which the reader can safely make the association to myelin content would be helpful in the introduction. Furthermore, many features are different between the myelin stains and MT profiles (Figure 1B). Some brief discussion of the differences would be helpful. Finally, it would be useful to clarify if to the author's knowledge this is the first time anyone has matched MT depth profiles to a histological measurement (or discuss relevant literature of past efforts).

The updated manuscript elaborated more on the relationship between MT, myelin, and the link between MT profiles and intracortical architecture. As suggested, we also extended our Discussion section on our novel comparison of MT profiles with myelin sections, specifically highlighting that differences in resolution and specificity to myelin produce differences in the profiles.

Updates were made to the Introduction, Results section and Materials and methods section:

“Several recent neuroimaging studies assessed intracortical microstructure in adolescence. One promising imaging technique is magnetisation transfer (MT), an MRI acquisition sequence that is sensitive to how water molecules interact with macromolecules in the brain, notably myelin (Schmierer et al., 2007). Although techniques such as MT cannot be equated with cortical myelin content per se (Serres et al., 2009a; Serres et al., 2009b), the MT parameter is dominated by myelin-related molecules making this technique a viable in vivo proxy for the contrast seen histologically in myelin basic protein (Koenig, 1991; Odrobina et al., 2005; Whitaker et al., 2016). A post mortem study in patients with multiple sclerosis has also shown that MT measures scale with demyelination and remyelination, suggesting dependence of this measure on myelin content (Schmierer et al., 2007).”

“Although post mortem myelin-stained sections and in vivo MT profiles differ in terms of resolution and specificity to myelin, we observed similar variations in mean and skewness of profiles across levels of laminar differentiation (Figure 1B), supporting the extension of profile analysis from histology to in vivo MT imaging.”

“The MT saturation parameter decouples the MT signal from the longitudinal relaxation rate, making it a semi-quantitative measure that is robust to relaxation times and field inhomogeneities (Weiskopf et al., 2013; Hagiwara et al., 2018).”

“To further understand architectural and cellular underpinnings of in vivo MT profiles, we assessed the similarity of the MT-derived profiles with intracortical myelin profiles computed from previously published post mortem myelin sections. While there are clear differences in the resolution and specificity to myelin between the histological staining and our in vivo proxy data, findings were overall supportive of a close association between in vivo MT profiles and those derived from myelin stained sections. This line of evidence supports the specificity of the MT features to intracortical myeloarchitecture and motivates efforts to increase the availability of post mortem histological measures for more detailed cross-validation studies of putative MRI markers of myelin.”

5) MT skewness and relative layer thickness. MT skewness may depend on the relative thickness of the different layers. It would be useful to clarify how much results explain the data beyond a simple cytoarchitectural analysis (e.g., are the depth profiles acting as a quantitative substitute for cytoarchitecture 'types', or is the inferred connection to myeloarchitecture relevant?) Furthermore, it would be useful to clarify if every node is squashed to the same absolute depth? This could impact interpretation: two areas with an identical ratio of myelination in deep relative to superficial layers would have different skewnesses if their middle layer was made thicker or thinner (thicker middle layer could push out the skewness). If this is the case, MT skewness could not simply be interpreted as a "ratio of myelination in deeper compared to more superficial layers"; it would more represent a measure of the preferential distribution of MT closer to the white matter boundary. Clarification is needed.

We thank the reviewers for this interesting question. We have carried out several precautions and additional analyses to dispel an interaction of thickness with MT profile results. First, please note that our MT sampling approach leveraged equivolumetric transformations for intracortical surface constructions (Waehnert et al., 2014), which accounted for variations in the laminar thickness with respect to curvature. Secondly, we have regressed out overall cortical thickness from the MT findings in Appendix-figure 2 and observed no major changes to the spatial pattern of MT moments or age related changes in MT moments when including thickness in the regression model. The revised paper furthermore tested whether laminar thickness had an effect on MT skewness. Specifically, we extracted laminar thickness from a recent segmentation of cortical layers in a post mortem volumetric reconstruction of a human brain (Wagstyl et al., 2019). We found that laminar thicknesses did not exhibit systematic relationships with skewness (-0.17<r<0.02). We have updated the Results section and added these findings into a new Appendix-figure 5:

“Furthermore, spatial correlation analysis suggested that MT moments were relatively independent of regional variations in laminar thickness derived from a post mortem volumetric reconstruction of a Merker-stained human brain (Wagstyl et al., 2019) (mean: -0.06<r<0.05. skewness: -0.17<r<0.02; Appendix—figure 5).”

6) Statistical testing. Every time a statistical analysis is performed (and a test statistic/p-value quoted), it should be clearly stated to the reader what it is. Non-normal distributions seem common in this data; Spearman's correlations may be appropriate. Furthermore, it appears that the spin-test based permutations are only used for some analyses but not others (e.g., only from the "Age-related changes… MT profiles" section onwards). Were spin tests used for earlier spatial association analyses? If not, can this be justified?

We thank the reviewers for this suggestion. In the updated manuscript, all spatial correlation analyses utilized Spearman correlations and corresponding p-values were determined via spin tests (AlexanderBloch et al., 2018). We have updated Figure 1 and Figure 4 to reflect the slight changes in statistics. The Results section and Materials and methods section have also been amended as follows:

“At a whole cortex-level, mean MT was highest in idiotypic cortex and decreased with less laminar differentiation, while skewness exhibited an opposite pattern (spatial correlation=-0.23, p_spin_<0.001; Figure 1C).”

“This sensory-fugal gradient reflects systematic variations in the MT profiles; it was strongly correlated with MT profile skewness (r=0.91, p_spin_<0.001; Figure 1D) and weakly with mean MT (r=0.19, p_spin_<0.001).”

“By relating age-related shifts in MPC to anchors with age-related changes in MT moments, we found that greater decreases in MT skewness were related to sensory anchor coupling (r=-0.63, p_spin_<0.001; Figure 4C). Conversely, restricted age-related skewness changes were associated with paralimbic anchor coupling (r=0.65, p_spin_<0.001).”

“We assessed the correspondence of the MPC gradient with the MT moments using node-wise Spearman correlations, with p-values determined against 10000 null models obtained from spin permutations (Alexander-Bloch et al., 2018). All spatial correlations were estimated and tested in the same way throughout the study.”

7) Utility of low-dimensional embedding. Regarding this statement: "Nodes closer in this embedding space increase in microstructural similarity during adolescence, whereas distant nodes decouple", it is unclear why a low-dimensional embedding is useful a direct measurement of change is available. Couldn't you more easily do an analysis directly (e.g., investigate the structure of the 'increasing' edges?) What is gained from this harder-to-interpret nonlinear embedding that helps test/interpret your hypotheses? Perhaps more importantly, it is unclear if this statement is supported by the data: Wouldn't the embedding put nodes close in the space that have similar "patterns" of age-related MPC change with other network nodes? In the plot of the behavior of G1_dev, it does not display the claimed behavior (Figure 4B) – instead, it seems like the largest values of G1_dev already have ~maximal G1_MPC, and thus barely change at all between the <16 and >24 groups. Clarification is warranted.

We agree with the reviewers’ recommendation that the G1_dev would be better described as “Nodes closer in this embedding space have more similar patterns of age-related change in MPC, whereas distant nodes undergo dissimilar development” (updated in the Results section). It follows from this definition that positioning of G1_dev reflects the principal axis of variance in developmental changes, but that does not necessarily correspond to changes in the values of G1_MPC.

Low-dimensional embedding was used to synoptically characterise coordinated age-related changes in MPC. We noted many significant age-related changes in MPC (1564 increases vs 534 decreases at q_FDR_<0.025), but they were not clearly circumscribed to a set of brain regions or level of laminar differentiation (Figure 4—figure supplement 1), therefore the low-dimensional embedding was helpful to visualise and interpret the patterns of change. In fact, the embedding approach led to the discovery that adolescent development expanded the existing sensory-fugal gradient of microstructural differentiation, and this would not have been easily observable when using the high-dimensional representations.

8) Biological interpretation. In the discussion, the biological interpretation of the results at times is rather thin. If MT is a strong myelin marker, then why would myeloarchitecture be reorganizing like this (e.g., to support what behavior)? Axonal myelination improves transmission speeds and prevents the formation of new synaptic connections. Could you speculate on how this might alter behavior relevant to the longitudinal changes analyzed? Relevant papers to consider include:*Micheva et al., (2016).*Braitenberg, (1962).*Huntenburg et al., (2017).

We have expanded the biological interpretation of our findings and provided further speculation on potential implications for behaviour, also citing the suggested papers.:

“Both sets of age-related changes were strongly and preferentially located in heteromodal and unimodal association cortex; yet, they only partially overlapped, a finding also supported by gene expression analysis. […] The architectural changes observed in the present cohort may represent a microstructural substrate that could determine overall cognitive and social capacities or susceptibility to a range of neuropsychiatric and neurological conditions occurring in late childhood and adolescence.”